# Suicide-Related Mortality Trends in Europe, 2012–2021

**DOI:** 10.3390/ijerph22060890

**Published:** 2025-06-02

**Authors:** Marco Zuin, Diego de Leo

**Affiliations:** 1Department of Translational Medicine, University of Ferrara, 44121 Ferrara, Italy; marco.zuin@edu.unife.it; 2Department of Cardio-Thoraco-Vascular Sciences and Public Health, University of Padova, 35137 Padua, Italy; 3Slovenian Centre for Suicide Research, Primorska University, 6000 Koper, Slovenia; 4Australian Institute for Suicide Research and Prevention, Griffith University, Brisbane 4122, Australia

**Keywords:** suicide, Europe, age, mortality rate, sex, older adults

## Abstract

Aims: Updated data regarding the suicide-related mortality trend in Europe remain scant. We assess the age- and sex-specific trends in suicide-related mortality in the European states (EU) between the years 2012 and 2021. Methods: We retrieved data on cause-specific deaths and population numbers by sex for European countries from the publicly available EUROSTAT mortality dataset for the years 2012–2021. This study was chosen because 2012 was the first year with complete uninterrupted suicide mortality data for all EU member states, while 2021 was the most recent year with confirmed estimates in the EUROSTAT database. Suicide-related deaths were identified using the International Classification of Diseases, 10th revision codes X60–X84 and Y870 as the underlying cause of death. We calculated annual trends by assessing the average annual percentage change (AAPC) with corresponding 95% confidence intervals (CIs) using joinpoint regression. Results: During the study period, there were 391,555 suicide-related deaths in Europe (313,835 men and 77,720 women). The age-adjusted mortality rate (AAMR) decreased linearly from 12.3 (95% CI: 12.0 to 12.6) per 100,000 people in 2012 to 10.2 (95% CI: 10.0 to 10.5) per 100,000 people in 2021 [AAPC: −2.3% (95% CI: −2.9 to −1.8); *p* < 0.001]. This decline was more pronounced among men [AAPC: −2.4% (95% CI: −2.9 to −2.0), *p* < 0.001] compared to women [AAPC: −1.9% (95% CI: −2.7 to −1.0), *p* < 0.001] (*p* for parallelism = 0.003). A more significant decrease was observed in individuals under 65 years compared to older individuals (*p* for parallelism = 0.001). Some EU subregions and demographic groups showed stagnation in suicide-related mortality rates. Conclusions: Over the past decade, age-adjusted suicide-related mortality has declined in Europe, particularly among males and individuals under 65 years old. However, disparities persist between countries and EU subregions.

## 1. Introduction

Suicide remains a significant public health challenge [1,2]. The World Health Organization has intensified efforts to promote prevention strategies by publishing documents, fact sheets, and reports, with notable examples including “*Preventing suicide: a global imperative*” and “*LIVE LIFE: an implementation guide for suicide prevention in countries*” [3,4,5]. To the best of our knowledge, there is a lack of recent studies analyzing trends in suicide within Europe. Globally, suicide has been remarkably declining during the last two decades [5]. However, the European region still presents the highest rate for male suicide [5]. It would be of interest to analyze what has happened in this region in the past decade (for which complete datasets are available) and see if notable phenomena/variations have occurred in the composition of trends among countries in terms of age and sex distribution. To address this knowledge gap, we analyzed data from the European Statistical Office (EUROSTAT) mortality dataset to assess current trends in European suicide-related mortality between the years 2012 and 2021 and determine differences by sex and age among European sub-regions.

## 2. Methods

### 2.1. Data Source

Data on suicide-related mortality and population numbers, without age restrictions, stratified by sex, and for European countries, were retrieved through the publicly available EUROSTAT mortality dataset for the years 2012–2021 [6]. The EUROSTAT mortality dataset provides information from death certificates of all European residents according to the International Classification of Diseases, 10th Revision (ICD-10). Specifically, this dataset collects data submitted by European states, including cause of death, age of death, sex of individuals, and population size stratified by sex. For the present analysis, suicide-related deaths were ascertained when ICD-10 codes X60–X84 and Y870 were listed as the primary cause of death in the medical death certificate, defined as the disease or medical condition which initiated the train of morbid events leading to deaths. These data are based on medically certified reports submitted annually to EUROSTAT by European states, which require legal certification of death as standard ICD codes. All the information transmitted by each member state is meticulously validated according to standard procedures, including a plausibility check, agreement with previous years, and validation against the ICD codes. Moreover, all the reports provided by different states undergo a standardized quality evaluation [7]. Specifically, if two or more underlying causes have been recorded on the form of the death certificate, the trigger that started the chain of events that led to death is chosen as the cause of death and classified according to the ICD-10 classification, as commonly performed in any other national or international mortality registry. The study period was chosen to start in the year 2012, which represents the first year without missing or data discontinuity on suicide-related mortality for all the EU states considered, whereas the year 2021 is, at the time of this study, the last year with confirmed and available suicide-related mortality data in the EUROSTAT dataset. This study did not require institutional review board approval since the analysis was based on de-identified and publicly available data. EUROSTAT is not responsible for the data elaboration presented in this analysis. It is noteworthy that the data reported by Eurostat correspond to “real data”, differing from the evidence provided by studies like the Global Burden of Disease, which generates estimates [8].

### 2.2. Data Extraction

Data extraction and validation were performed separately by two independent investigators (MZ, DDL). Age-adjusted mortality rates (AAMRs) per 100,000 people were computed for each country by sex, standardizing aggregate rates to the European standard population (ESP) [9], using the direct method to allow for a comparison both over time and among countries independently from the population age and structures. The EU states included in the analysis were Austria, Belgium, Bulgaria, Croatia, Cyprus, Czech Republic, Denmark, Estonia, Finland, France, Germany, Greece, Hungary, Ireland, Italy, Latvia, Liechtenstein, Lithuania, Luxembourg, Malta, the Netherlands, Norway, Poland, Portugal, Romania, Serbia, Sweden, Slovakia, Slovenia, Spain, and Switzerland.

### 2.3. Statistical Analysis

The different geographical distribution of suicide-related mortality was reported as AAMRs per 100,000 people over the entire study period. To report the annual trends in suicide-related mortality, we assessed the annual percent change (APC), the average annual percent change (AAPC), and the relative 95% confidence intervals (CIs) using joinpoint regression analysis. This statistical method was chosen since it offers several advantages over traditional linear trend analysis, including the detection of significant changes, or “joinpoints”, in the trend of a time series dataset, as well as the identification of periods of acceleration or deceleration in the trend, which may be missed by other types of time trend analysis as linear trend analysis. A parallelism test (pairwise comparison) was used to compare different sets of trends with mean functions estimated by the joinpoint regression analysis [10]. Specifically, a significant *p*-value in this interaction test indicated that the two trends, in terms of AAPC, were statistically significantly different from each other. Proportional mortality was defined as the ratio between the number of suicide-related deaths and 100 deaths for all causes. Moreover, specific sub-analyses by sex, age, and EU subregions were also performed. Statistical significance was pre-specified at *p* ≤ 0.05 for findings in the entire population. Statistical analyses were performed using joinpoint regression (Joinpoint, version 4.6.0.0, National Cancer Institute, Bethesda, Washington, DC, USA).

## 3. Results

### 3.1. Overall EU Population

From 2012 to 2021, a total of 46,891,752 individuals (23,410,516 men and 23,481,236 women) died in the considered EU countries. Suicide was listed as the underlying cause in 391,555 patients (313,835 men and 77,720 women), equating to 835 deaths per 100,000 individuals in the period. The AAMR for suicide-related mortality linearly decreased, without any inflection points, from 12.3 (95% CI: 12.0 to 12.6) per 100,000 people in 2012 to 10.2 (95 CI: 10.0 to 10.5) per 100,000 people in 2021 [AAPC: −2.3% (95% CI: −2.9 to −1.8; *p* < 0.001]; (Table 1, Figure 1 and Figure 2 and Appendix A).

### 3.2. Proportionate Mortality

The proportionate mortality declined from 0.98% to 0.68% (*p* for trend 0.001), with a similar trend among sexes (from 1.60% to 1.08%, *p* for trend 0.001 in men; and from 0.37% to 0.27%, *p* for trend 0.01 in women, respectively) (Figure 3).

### 3.3. Sex

Over the entire study period, men had greater suicide-related mortality and a greater decrease in the AAMR compared with women (*p* for parallelism = 0.003). In men, the AAMR decreased from 20.7 (95% CI: 20.2 to 21.1) per 100,000 people in 2012 to 16.7 (95% CI: 16.5 to 16.9) per 100,000 people in 2021 (AAPC: −2.4%; 95% CI: −2.9 to −2.0; *p* < 0.001). Specifically, men showed a biphasic trend, since the AAMR linearly decreased from 2011 to 2017 [APC: −3.4%, (95% CI: −5.4 to −2.7), *p* < 0.001] and then plateaued from 2017 to 2021 [APC: −1.0 (95% CI: −2.3 to 1.7), *p* = 0.33]. In women, the AAMR decreased, without any inflection points, from 5.2 (95% CI: 5.0 to 5.5) per 100,000 people in 2012 to 4.5 (95% CI: 4.3 to 4.7) per 100,000 people in 2021 (AAPC: −1.9%; 95% CI: −2.7 to −1.0; *p* < 0.001) (Table 1 and Figure 1).

### 3.4. Age

The suicide mortality rate increased with aging, reaching a peak between 50 and 54 years and then decreasing in older age groups (Appendix A). From 2010 to 2019, a total of 298,950 individuals <65 years old (263,280 men and 35,670 women) died due to suicide. This demographic group demonstrated a greater decrease in suicide-related mortality compared with patients aged 65 years or older throughout the study period (*p* for parallelism = 0.001). Specifically, among younger patients, suicide-related mortality decreased from 9.1 (95% CI: 8.8 to 9.4) per 100,000 in 2012 to 8.6 (95% CI: 8.3 to 9.0) per 100,000 in 2021 (AAPC −2.7%;95% CI: −3.0 to −2.3; *p* < 0.001). In contrast, among subjects aged 65 years and older (92,605 subjects, 50,555 men and 42.050 women), the AAMR decreased from 2012 to 2019 [APC: −2.2 (95% CI: −3.8 to −1.7), *p* = 0.003] and then plateaued from 2019 to 2021 [APC: 1.9% (95% CI: −1.2 to 3.7, *p* = 0.30) (Table 1 and Appendix A).

### 3.5. Trends in Suicide-Related Mortality by European Sub-Regions

Suicide-related AAMRs decreased across Europe, except in northern countries where they plateaued [AAPC: −0.8% (95% CI: −1.7 to 0.1), *p* = 0.06]. Sub-analysis by gender showed consistent declines in men but plateauing in women in the northern countries [AAPC: −0.2% (95% CI: −2.4 to 2.2), *p* = 0.85] and southern EU [AAPC: −1.5% (95% CI: −3.2 to 0.2), *p* = 0.08]. Age stratification indicated stagnation in suicide-related mortality for individuals under 65 and aged 65 years and more in northern countries [AAPC: −0.8% (95% CI: −1.9 to 0.3), *p* = 0.13; AAPC: −2.8% (95% CI: −6.6 to 1.1), *p* = 0.13], as well as those over 65 in western EU [AAPC: 0.1% (95% CI: −2.0 to 2.2), *p* = 0.99]. The highest AAMR decrease (AAPC ≥ −4%) was observed in Cyprus, Latvia, Lithuania, and Hungary (Table 2 and Appendix A).

## 4. Discussion

This analysis, based on medically certified vital registration data from the EUROSTAT mortality dataset, provides updated estimates regarding the suicide-related mortality trends, covering a population of 32 European countries. Present findings revealed that suicide represented about 1% of the total deaths over a decade in Europe, confirming its importance as a cause of death in the European population, especially among men and individuals aged less than 65 years. Although the suicide-related mortality has declined in most of the European countries, a stagnation was observed in some EU subregions and across some demographic groups. Of some relevance is the observation that, overall, during the first two years of the COVID-19 pandemic (2020–2021), suicide did not increased in Europe.

Despite the reduction in suicide-related AAMRs, in the proportionate mortality, men continue to bear a disproportionately higher burden of suicide-related mortality, accounting for approximately 76% of all deaths by suicide [11,12]. Interestingly, while men showed a biphasic trend with a plateau from 2017 to 2021, women exhibited a linear decline without inflection points. These findings underscore the need for targeted interventions addressing gender-specific risk factors and prevention strategies [13]. Age stratification revealed that suicide-related mortality peaked among individuals aged 50–54 years and subsequently declined in older age groups. Notably, among older adults, the decline plateaued after 2019, possibly reflecting unique challenges such as social isolation or health-related vulnerabilities in this demographic [14]. These observations highlight the importance of age-tailored prevention strategies to address the distinct needs of different age groups [15,16,17]. Attributing a precise motive to the decline in suicide rates remains highly speculative. It is uncertain if suicide prevention strategies had a role in this phenomenon, which crossed into the time of the pandemic, a period that could potentially trigger more suicide cases for its many challenges (e.g., economic, emotional, social, etc.). Also, the presented geographical heterogeneity in suicide-related mortality trends across European sub-regions deserves some consideration. Indeed, these differences need further clarifications, although it could be possible that local policies, healthcare preventive campaigns, and access to healthcare systems and therapeutic interventions may play/have played a critical role in shaping outcomes [18]. Throughout the study period, ameliorated access to health services and an improved quality of life of Europeans could have had a role. However, socio-economic-oriented research could eventually provide support for this hypothesis.

Suicide rates have been declining in Europe during the most recent decade for which data are available (2012–2021). Even if the most recent global rate is higher (10.2/100,000) than the one reported by the WHO (9.0/100,000) [19], the decline was significant (from 12.3/100,000 to 10.2/100,000, approximately a 17% decline). Compared to our data, which are derived from the EUROSTAT mortality dataset, previous estimates on suicide-related mortality based on the GBD dataset have several limitations [19]. GBD relies on modeling techniques, such as the Death Ensemble model, to estimate suicide mortality, often correcting for underreporting and misclassification through the redistribution of ill-defined ICD codes. While this approach enhances comparability across regions, it introduces uncertainty due to reliance on indirect estimates and assumptions, particularly in areas with limited or poor-quality data [20]. In contrast, EUROSTAT data are based directly on death certificates, representing “real deaths” without the need for modeled adjustments. This ensures higher accuracy in reflecting actual mortality trends but may still be influenced by variations in national reporting practices and a potential underreporting of suicides due to stigma or cultural factors.

The decline in suicide rates and the marked regional heterogeneity observed across Europe between 2012 and 2021 likely reflect the combined impact of several interrelated factors. First, the shift from hospital-centered to community-oriented care has improved early identification, continuity of care, and social reintegration for people with psychiatric disorders. In this regard, it has been reported that regions that invested in mobile teams, drop-in centers, and assertive outreach saw steeper declines in suicide mortality, suggesting that ready access to psychosocial support is a key protective factor [20]. Second, the implementation of socioeconomic support and welfare policies mitigate stress, reinforcing social safety downstream in mental health consequences. To this regard, a systematic review found that, paradoxically, in Spain, each 1 % rise in male unemployment was associated with a decrease (RR = 0.84) in suicide-related healthcare contacts, likely reflecting the increased uptake of generalist health services under crisis relief programs [21]. Moreover, local community structures played a crucial role in influencing suicide risk. In the Netherlands, a population-based case–control study found that people living in socially fragmented or poor neighborhoods faced twice the suicide mortality rate of those in more cohesive communities, regardless of their individual socioeconomic status [22], suggesting that peer support groups could help reduce suicide rates at the population level. Additionally, targeted interventions for high-risk demographics have probably influenced our observed trends. A multicenter trial combining primary care gatekeeper training, loneliness reduction programs, and telepsychiatry for rural elders achieved a 12% reduction in suicide ideation and a 7% drop in deaths by suicide over five years [23]. These targeted approaches illustrate how demographically tailored strategies can reinforce broader suicide prevention frameworks.

Although suicide rates did not rise during the initial phase of the COVID-19 pandemic, several converging influences may help explain this unexpected stability. First, the rapid expansion of telehealth services ensured the continuity of mental healthcare when in-person visits were limited; one review reported a three- to five-fold increase in telepsychiatry encounters, which likely mitigated treatment disruptions [24]. Second, comprehensive economic relief measures, such as furlough schemes and direct cash transfers across many European countries, helped buffer financial stress, a known suicide precipitant [25]. Finally, heightened public and community attention to mental well-being, including widespread media campaigns and grassroots mutual aid networks, may have fostered social connectedness and early help-seeking behaviors [26]. Together, these factors offer a plausible explanation for why suicide mortality remained stable despite the profound social upheaval of the pandemic.

The marked decline in suicide rates across Europe has been facilitated in part by the “Optimizing Suicide Prevention Programmes and their Implementation in Europe” (OSPI Europe) initiative. Central to OSPI’s success were formal partnerships among health services, non-governmental organizations, and local government bodies, which enabled coordinated, multi-sectoral response efforts [27]. Moreover, each participating country adapted the core intervention components—public awareness campaigns, gatekeeper training, media guidelines, and enhanced support for high-risk individuals—to fit local cultural contexts and healthcare infrastructures. Regular process evaluation checkpoints, guided by predefined fidelity metrics, allowed implementers to troubleshoot in real time and ensure that key activities (e.g., public presentations, media engagement) reached their target audiences. Notably, regions that sustained over 80% fidelity across all intervention components experienced the largest decreases in both suicidal ideation and suicide-related service utilization [28].

The present results carry several actionable lessons for public health policy: regions and subgroups with an elevated suicide burden (e.g., elderly men in eastern Europe) should receive proportionally greater funding for mental health services and outreach programs. Economic modeling indicates that prioritizing high-risk groups can yield the greatest reductions in suicide mortality per euro invested [29]. Furthermore, national strategies should incorporate interventions proven at the population level, such as restricting access to lethal means, school- and workplace-based programs, and media reporting guidelines. A decade-long systematic review demonstrated that combining these measures leads to an average 20–30% decline in suicide rates across diverse settings [30]. Additionally, our regional analysis supports the expansion of community gatekeeper training—equipping teachers, employers, and primary care providers to identify and refer at-risk individuals. Cost-effectiveness studies show that gatekeeper programs reduce emergency presentations for self-harm by up to 15% when integrated into existing health infrastructures [31]. Finally, establishing real-time surveillance systems and linking coronial, hospital, and social service data will allow policymakers to detect emerging hotspots and evaluate intervention impact. Countries that have adopted such systems report 10–12% greater accuracy in identifying high-risk demographics and locales, enabling an agile reallocation of resources [30].

### Limitations

Our study has several important limitations. First, as with any research relying on a large administrative mortality dataset, we cannot rule out the potential for miscoding and/or misdiagnosis, which may have affected the accuracy of our analysis. While a direct validation of the various ICD-10 death codes used to define suicide-related death has not yet been conducted using the EUROSTAT dataset, previous analyses employing the same ICD-10 codes have demonstrated acceptable accuracy and positive predictive values, ranging from 88% up to 100% [32,33,34,35]. Additionally, because the EUROSTAT dataset is a mortality registry, it does not include data on prior mental health disease, risk factors, comorbidities, treatment, or information regarding potential competing causes of death, which further limits our analysis. Accurate suicide surveillance is crucial for effective prevention, yet persistent underreporting and misclassification can substantially distort the true burden of suicide. However, in different countries, societal stigma and legal disincentives, such as the implication of insurance disputes or family dishonor, lead coroners and registrars to attribute ambiguous deaths to “accidental” or “undetermined” causes rather than suicide. Luoma et al. estimated that up to 20% of actual suicides may be hidden within undetermined intent categories in vital statistics, underlining how cultural pressures skew official figures [36]. Moreover, the criteria and thoroughness applied by medical examiners vary widely between and within countries. Rockett et al. demonstrated that regions with limited access to forensic autopsies report consistently lower suicide rates, suggesting that procedural gaps contribute to systematic undercounting [37]. Underreported suicides impede the identification of high-risk populations and the evaluation of prevention strategies. They also compromise international comparisons: countries with rigorous certification systems appear to have higher suicide rates than those with lax reporting, paradoxically penalizing better performing health systems. Without adjustment for underreporting, resource allocation may be misdirected away from areas of true need.

## 5. Conclusions

This study highlights a significant decline in suicide-related mortality rates across Europe between 2012 and 2021, with the age-adjusted mortality rate (AAMR) decreasing by approximately 17%. The reduction was more pronounced among individuals under 65 years of age (−20%) and slightly greater in men compared to women, although men continue to account for the most suicides. However, the most recent European suicide rate remains higher than the global average reported by the WHO. Attributing this decline to specific factors remains speculative. While improved access to healthcare services and enhanced quality of life in Europe may be contributors, it is uncertain whether suicide prevention strategies played a decisive role, particularly given the challenges posed by the COVID-19 pandemic. This period, marked by economic, emotional, and social stressors, could have potentially increased suicide risk but did not reverse the declining trend. Further socio-economic research is needed to better understand the drivers of this positive development and to identify effective strategies for sustaining and accelerating reductions in suicide-related mortality across diverse populations.

## Figures and Tables

**Figure 1 ijerph-22-00890-f001:**
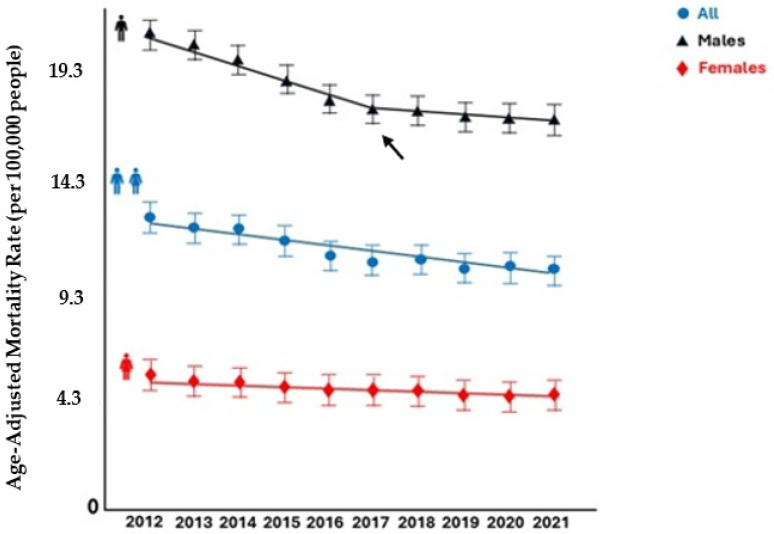
Suicide-related mortality rates in Europe by sex, 2012–2021. The arrow indicates a joinpoint.

**Figure 2 ijerph-22-00890-f002:**
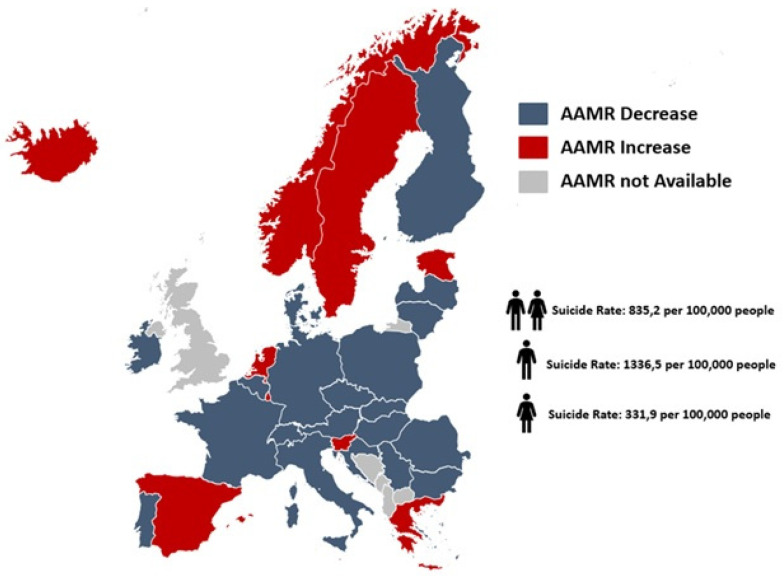
Map of trends in suicide-related age-adjusted mortality rates among European countries, 2012–2021.

**Figure 3 ijerph-22-00890-f003:**
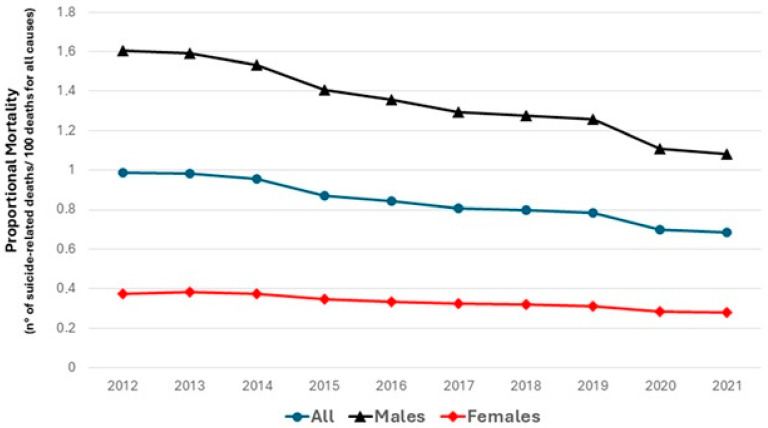
Proportional mortality for suicide-related mortality in Europe between 2012 and 2021, stratified by sex.

**Table 1 ijerph-22-00890-t001:** Suicide-related age-adjusted mortality in Europe, stratified by sex and age, for the entire population, 2012–2020.

	**Age-Adjusted Mortality Rate (per 100,000)**															
	**2012**	**2013**	**2014**	**2015**	**2016**	**2017**	**2018**	**2019**	**2020**	**2021**	**AAPC**	**95% CI**	** *p* **	**Joinpoints**	**APC** **Period 1** **(Years);** **APC** **(95% CI);** ** *p* **	**APC** **Period 2** **(Years);** **APC** **(95% CI);** ** *p* **
Europe	12.3	12.2	11.8	11.3	10.7	10.4	10.3	10.1	10.2	10.2	−2.2	−2.7 to −1.8	<0.001	0	-	-
Sex																
Europe Men	20.7	20.6	20.2	19.5	18.7	17.7	17.2	16.8	16.7	16.6	−2.4	−2.9 to −2.0	<0.001	1	[2012–2017] −3.4 (−5.4 to −2.7) *p* < 0.001	[2017–2021] −1.0 (−2.3 to 1.7) *p* = 0.33
Europe Women	5.2	5.2	5.3	5.1	5.0	4.6	4.6	4.6	4.4	4.3	−1.9	−2.7 to −1.0	<0.001	0	-	-
Age																
<65 years	10.8	10.6	10.2	9.7	9.2	9	8.9	8.7	8.7	8.6	−2.7	−3.0 to −2.3	<0.001	0	-	-
≥65 years	18.8	18.7	18.2	17.9	17.0	16.6	16.9	16.2	16.5	16.8	−1.3	−1.9 to −0.9	<0.001	1	[2012–2019] −2.2 (−3.8 to −1.7) *p* = 0.003	[2019–2021] 1.9 (−1.2 to 3.7) *p* = 0.30

AAMR: age-adjusted mortality rate, expressed as deaths per 100,000 people. AAPC: average annual percent change. APC: annual percent change.

**Table 2 ijerph-22-00890-t002:** Suicide-related age-adjusted mortality in Europe, stratified by European subregions, exploring different trends in the general population, 2012–2020.

	Age-Adjusted Mortality Rate (per 100,000)																
	2012	2013	2014	2015	2016	2017	2018	2019	2020	2021	AAPC	95% CI	*p*	Joinpoints	APC Period 1 (Years); APC (95% CI); *p*	APC Period 2 (Years); APC (95% CI); *p*	P for Parallelism
All																	North vs West *p* = 0.06 North vs West *p* = 0.004 North vs South *p* = 0.003 West vs East *p* = 0.01 West vs South *p* = 0.55 East vs South *p* = 0.04
North	12.6	14.4	12.9	12.6	12.6	12.0	12.5	12.4	12.3	11.8	−0.8	−1.7 to 0.1	0.06	0	-	-	
West	15.5	15.1	15.0	13.9	13.8	13.8	13.6	13.1	12.9	11.4	−2.6	−3.6 to −1.6	<0.001	0	-	-	
East	18.3	18.1	17.3	16.2	15.2	15.2	14.0	13.2	13.1	12.3	−3.8	−4.7 to −3.0	<0.001	1	(2012–2013) 0.1 (−3.9 to 4.6) *p* = 0.72	(2013–2021) −4.7 (−8.0 to −4.0) *p* < 0.001	
South	9.6	9.7	9.6	9.4	8.4	8.3	8.2	8.1	8.0	7.9	−2.0	−3.7 to −0.3	0.02	0	-	-	

AAMR: age-adjusted mortality rate, expressed as deaths per 100,000 people. AAPC: average annual percent change; APC: annual percent change/southern Europe: Greece, Spain, Cyprus, Malta, Portugal, Slovenia, and Serbia; northern Europe: Finland, Sweden, Iceland, and Norway; western Europe: Belgium, Denmark, Germany, Ireland, France, Luxemburg, Hungary, the Netherlands, Austria, Poland, Lichenstein, and Switzerland; eastern Europe: Bulgaria, Czech Republic, Estonia, Latvia, Lithuania, and Slovakia.

## Data Availability

The original contributions presented in this study are included in the article and Appendix A. Further inquiries can be directed to the corresponding author.

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
