# Peer review of "Suicide-Related Mortality Trends in Europe, 2012–2021"

_ijerph, 2025, doi:10.3390/ijerph22060890_

Round 1
Reviewer 1 Report
Comments and Suggestions for Authors
This is an academic text that addresses one of the public health problems considered to be of the first order in European countries. The text notes, based on the statistical analysis of national figures provided by Eurostat, the overall decrease in the number of suicides between 2012 and 2021, with significant variations by age group, sex and subregions.
The conclusions are relevant at the descriptive level, although more depth in theoretical reflection would be desirable.
The manuscript is clear and well-structured.
Regarding the Methodology:
The Data Source is well-delimited and clearly stated.
The Data Extraction and Statistical Analysis are correct. Appropriate methodologies are used. The use of a parallelism test is particularly interesting for trend comparison.
About the results:
The results are correctly stated. The graphs are relevant and visually support the content of the text.
About the discussion and conclusions:
Expanding the reflections around the problems caused by underreporting is recommended.
The limitations are well specified.
In these sections, the text is too descriptive, and there is no in-depth approach to the reasons for the decrease in suicides nor to the geographical variations. Although in the article, the authors point out that putting forward hypotheses in this sense is too speculative, there is sufficient scientific literature, particularly at the national level, which, based on evidence, provides explanatory proposals on the reasons for the suicide figures. It is recommended that the authors expand these sections in this sense.
Regarding the bibliography:
This is adequate for the descriptive purpose of the article. Still, it is recommended to expand it with texts that facilitate understanding the identified phenomena: the reduction of suicides.
These two bibliographic references are undated:
- World Health Organization. Preventing suicide: a global imperative. WHO, Geneva. ISBN 9789241564779 273
- World Health Organization. LIVE LIFE: An implementation guide for suicide prevention in countries. WHO, Geneva ISBN: 274 9789240026629
It is also recommended to search for scientific literature on Optimizing suicide prevention programs and their implementation in Europe (OSPI Europe).
Author Response
Please find authors' responses in the attached file. Thank you.

Reviewer 2 Report
Comments and Suggestions for Authors
This manuscript presents a valuable and timely analysis of suicide-related mortality trends in Europe from 2012 to 2021, using reliable EUROSTAT data and robust statistical methods. The findings offer important insights into demographic and regional patterns, including notable declines in overall suicide rates, particularly among men and those under 65. The analysis is well-executed and contributes meaningfully to the literature on suicide epidemiology in high-income countries.
However, several revisions are needed to improve clarity, consistency, and contextual interpretation. See below for details.
Clarity of Objectives
The abstract and introduction clearly describe the purpose of the study, which is to analyze suicide-related mortality trends in Europe by age and sex. However, the rationale for selecting the 2012–2021 timeframe is not immediately apparent. Although the methods section notes that 2012 was the first year without missing data across countries, this explanation should be briefly mentioned in the abstract or introduction to provide clearer context upfront.
COVID-19 Context
While the authors note that suicide rates did not increase during the early years of the COVID-19 pandemic (2020–2021), the discussion could be strengthened by offering more interpretation of this finding. Briefly discussing potential influences—such as expanded telehealth services, economic relief policies, or increased community attention to mental health—would enhance the manuscript’s insight into this unexpected trend.
Statistical Presentation
The use of Joinpoint regression is appropriate and effectively applied. However, the manuscript would benefit from clearer visual presentation of the statistical results For instance, the manuscript notes a "biphasic trend" in men's suicide-related mortality rates: a significant decline from 2011 to 2017, followed by a plateau from 2017 to 2021. This suggests an inflection point around 2017, where the rate of decline slowed or stopped. To enhance clarity, it's beneficial to visually represent these inflection points in your figures. By clearly marking the year where the trend changes—such as 2017 in this case—readers can more easily grasp the shifts in the data.
Underlying reasons
The authors note that “attributing precise motive to the decline in suicide rates remains highly speculative.” However, offering informed speculations grounded in existing literature would enhance the paper’s contribution. I recommend that the authors propose potential explanations for each of the major trends identified, drawing on relevant empirical or theoretical frameworks where possible.
Language and Style
There are a few minor grammatical issues and unclear phrases that detract slightly from the overall readability of the manuscript. For example, the statement “equating to 835.2 deaths per 100,000” might be an error and need to be double checked. A thorough proofreading pass would help address such instances and improve overall flow.
Policy Implications
The discussion section does a good job summarizing demographic and regional findings but stops short of considering their implications for public health policy. A brief mention of how the results might inform future prevention strategies or resource allocation, especially for high-risk subgroups, would enhance the practical significance of the study.
Limitations
The authors should also consider discussing the likelihood of underreporting suicide deaths due to cultural stigma or inconsistent death classification practices across countries, which may bias cross-national comparisons.
Author Response

(The authors gave the same response as above.)
